# The Associations between Upper and Lower Body Muscle Strength and Diabetes among Midlife Women

**DOI:** 10.3390/ijerph192013654

**Published:** 2022-10-21

**Authors:** Beverly W. X. Wong, Win Pa Pa Thu, Yiong Huak Chan, Michael S. Kramer, Susan Logan, Jane A. Cauley, Eu-Leong Yong

**Affiliations:** 1Department of Obstetrics and Gynecology, National University Hospital, National University of Singapore, Singapore 119228, Singapore; 2Biostatistics Unit, Yong Loo Lin School of Medicine, National University of Singapore, Singapore 117597, Singapore; 3Department of Epidemiology and Biostatistics and of Pediatrics, Faculty of Medicine, McGill University, Montreal, QC H3G 2M1, Canada; 4Department of Epidemiology, Graduate School of Public Health, University of Pittsburgh, Pittsburgh, PA 15261, USA

**Keywords:** muscle strength, diabetes, midlife, Singapore, IWHP

## Abstract

We hypothesized that a combined index of upper and lower body muscle strength would be more strongly associated with diabetes than either measure alone. Women recruited into the Integrated Women’s Health Program had their handgrip strength (HGS) measured using a dynamometer and underwent a timed 5-repetition chair stand (RCS) test. HGS < 18 kg and RCS performance ≥ 12 s assessed upper and lower body strength, respectively, both individually and combined in a muscle strength index (MSI). Diabetes was defined as physician-diagnosed, use of anti-diabetic medication, or fasting blood glucose ≥ 7.0 mmol/L. Binary logistic regression examined the associations between muscle strength and diabetes. Of 1170 midlife women, 12.1% had diabetes. A low HGS was independently associated with diabetes (aOR: 1.59, 95% CI: 1.03, 2.44). Prolonged RCS was also associated with diabetes (aOR: 1.59, 95% CI: 1.09, 2.30), but this was not independent of visceral adiposity and muscle mass. A poor MSI had higher odds of diabetes (aOR: 2.37, 95% CI: 1.40, 4.03), independent of age, ethnicity, education level, menopausal status, smoking, alcohol consumption, physical activity, height, visceral adiposity, and muscle mass. The combination of both upper and lower body muscle strength into a composite MSI was more strongly associated with diabetes than either weak HGS or prolonged RCS alone in midlife women.

## 1. Introduction

Diabetes has strong associations with increased risks for mortality and morbidity, and is hence an important public health concern [1]. Globally, the number of adults with diabetes quadrupled from 108 million in 1980 to 422 million in 2014 [1]. Over 60% of individuals with diabetes are predicted to come from Asia, the world’s most populous region [2]. In Singapore, one in every five adults are predicted to have diabetes by 2035 [3]. Although established factors such as older age, obesity, reduced education, smoking, and physical activity explain a substantial proportion of diabetes risk [4,5], an important fraction of residual associations remain unexplained.

Muscle plays an important role in glucose metabolism, and previous studies have reported an association between diabetes and sarcopenia [6,7]. The chief determinant of sarcopenia is a shift from muscle mass to strength, as muscle strength predicts unfavorable health outcomes better than muscle mass [8]. Improving muscle strength through resistance training has been shown to enhance glycemic control in patients with diabetes [9,10]. Muscle strength decline is more prominent among women than men after 55 years of age [11], possibly due to reduced muscle protein synthesis during the menopausal transition period [12]. Few studies have examined the contribution of declining muscle strength to diabetes in midlife women [10,13].

Handgrip strength (HGS) is a convenient and reproducible method used to measure upper body muscle strength and is commonly regarded as a proxy for overall muscle strength [13]. Reduced HGS has been associated with increased risks of cardiovascular disease and all-cause mortality [14], and with respiratory diseases and breast cancer in the UK Biobank study [15]. However, the association between grip strength and adult-onset diabetes remains unclear. Few studies have related muscle strength to diabetes in Asian populations, with two recent systematic reviews including only one and two Asian cohorts out of ten and thirteen eligible cohorts, respectively [10,13].

Several studies have cautioned against the use of HGS as the sole measure of muscle strength [16,17]. A study among community-dwelling older women revealed a poor correlation between HGS and variables assessed by an isokinetic dynamometer, the gold standard for evaluating muscle function [16]. Although the lower body constitutes ~75% of appendicular lean mass [18], the relationship between lower body muscle strength and diabetes has been relatively unexplored. Precise measurement of lower limb strength in clinical settings has been limited as the equipment required is costly and non-portable [19]. A simple and cost-effective method to measure lower limb strength is the timed 5-repetition chair stand (RCS) test [20]. Studies examining the relationship between RCS and diabetes are sparse, with conflicting results, and were mainly conducted among Caucasian populations [21,22,23], with data rarely reported among Asian women.

These knowledge gaps led us to examine the relationship of both upper and lower body strength with diabetes in a cohort of midlife Singaporean women. We also tested the hypothesis that an index combining both upper and lower body muscle strength, as a novel proxy for whole-body muscle strength, would be more strongly associated with diabetes than either measure alone [17].

## 2. Materials and Methods

### 2.1. Study Design and Participants

The Integrated Women’s Health Program (IWHP) is a cross-sectional study examining health conditions among Chinese, Malay, and Indian midlife women, aged 45–69 years [24]. In brief, healthy participants attending well-women clinics were recruited from the National University Hospital, Singapore. Out of the 2715 women screened, 1201 women (54.8% participation rate) were enrolled into the study. Exclusion criteria included pregnancy or a potentially life-threatening health condition. Further details on the inclusion and exclusion criteria have been published previously [24]. This study was approved by the Domain Specific Review Board of the National Healthcare Group, Singapore (reference number: 2014/00356), and all participants provided written informed consent.

### 2.2. Handgrip Strength (HGS) and 5-Repetition Chair Stand Test (RCS)

HGS was measured using a hand-held hydraulic dynamometer (Jamar, Bolingbrook, IL, USA) [24]. The Jamar dynamometer is widely used to assess grip strength and is the gold standard by which other dynamometers are evaluated [25,26]. It has demonstrated high test-retest, inter-rater, and intra-rater reproducibility [25] and validity [26]. Participants held the dynamometer while seated with their forearm resting on a table and parallel to the ground, with the elbow flexed at a 90° angle. Participants were instructed to squeeze as hard as they could for approximately three seconds, and two readings were recorded from each arm by trained study personnel. Unlike previous studies in the IWHP cohort, which examined average grip strength over the four readings, we used the maximum reading recorded instead, as now recommended by the Asian Working Group for Sarcopenia 2019 [27]. Participants were classified as having low upper body muscle strength when HGS < 18 kg [27].

The RCS test Is widely used as a measure of assessing lower limb strength and was performed as part of the Short Physical Performance Battery [28]. A systematic review of 10 articles confirmed the high test-retest reproducibility of the RCS in the populations studied [29]. The RCS test has been shown to be a valid instrument of functional mobility and dynamic balance [30]. Trained study personnel instructed participants to firstly perform a single chair stand test with both arms folded across the chest, standing up when they heard the word “Go.” One successful chair stand involved rising to a full standing position from a seated position and sitting all the way back down again. Upon successful completion of the single chair stand test, participants were then asked to stand up five times as quickly as they could with both arms folded across the chest. Trained study coordinators demonstrated both the single chair stand and RCS before participants were assessed. The five-repetition was conducted once, with the time taken to complete it recorded in seconds. An RCS test requiring ≥12 s was classified as prolonged [27].

### 2.3. Muscle Strength Index (MSI)

To evaluate our hypothesis that a proxy for whole-body muscle strength would be better associated with diabetes than either HGS or RCS alone, we used cut-off values for both HGS and RCS from the Asian Working Group for Sarcopenia to form a novel composite measure, the muscle strength index (MSI). Participants with both HGS < 18 kg and RCS ≥ 12 s were classified as having “poor” muscle strength; those with either HGS < 18 kg only or RCS ≥ 12 s only were classified as having “intermediate” muscle strength while participants with both HGS ≥ 18 kg and RCS < 12 s were classified as having “normal” muscle strength.

### 2.4. Definition of Diabetes Status

Blood samples were drawn following an overnight fast, processed within 6 h, and stored at −80 °C until assayed. Fasting blood glucose levels were analyzed at the National University Hospital Referral Laboratory using a photometric hexokinase method assay (Beckman Coulter, Inc., Brea, CA, USA) [31]. The concentration range was 2.1–24.4 mmol/L, and the intra-assay and inter-assay coefficients of variance were 0.6–1.6% and 0.8–2.0%, respectively.

Diabetes was defined as any of the following: (1) self-reported physician-diagnosed diabetes; (2) self-reported use of anti-diabetic medication (insulin or oral medications); or (3) fasting blood glucose ≥7.0 mmol/L, in accordance with the American Diabetes Association criteria [32]. Because we did not inquire about type 1 or type 2 diabetes nor their age at onset, on the baseline questionnaire, our outcome was defined only as “diabetes.”

### 2.5. Sociodemographic, Reproductive, and Lifestyle Factors

Information on sociodemographic, reproductive, and lifestyle factors was self-reported using validated questionnaires, as described previously [24]. These factors included age, ethnicity, education level, housing type, parity, menopausal status, smoking status, and alcohol consumption. Physical activity was defined as having spent ≥ 150 min per week on moderate intensity physical activity and/or ≥75 min per week on vigorous intensity physical activity, measured by the Global Physical Activity Questionnaire [33]. Height was measured twice using a stadiometer, with the average value calculated and recorded, while weight was measured once using an electronic weighing scale (SECA 769). Participants were instructed to remove their shoes and empty their pockets, including their keys, wallets, and handphone. Body mass index (BMI) was calculated by taking height (in meters) divided by weight squared (kg^2^). According to the cutoff for Asian populations [34], underweight was defined as BMI < 18.5 kg/m^2^, overweight as ≥23 kg/m^2^, and obese as >27 kg/m^2^. Skeletal muscle mass in the form of appendicular lean mass (ALM) and visceral adipose tissue (VAT) in cm^2^ were obtained via dual-energy X-ray absorptiometry. ALM was normalized to ALM/height^2^, as recommended by the Asian Working Group for Sarcopenia [27]. VAT was categorized into tertiles, as lowest (<88.6 cm^2^), middle (88.6 to 131 cm^2^), and highest (>131 cm^2^).

### 2.6. Statistical Analysis

All analyses were performed using SPSS version 27.0 (IBM Corp., Armonk, NY, USA), with statistical significance set at 2-sided *p* < 0.05. Differences in participant characteristics, presented as n (%), between patients with and without diabetes were compared using Pearson’s chi-squared test. Hierarchical binary logistic regression analyses were performed to assess the associations between HGS, RCS, MSI, and diabetes. Binary logistic regression is a commonly and widely used statistical technique, in which the dependent variable is binary and the independent variables can be continuous and/or categorical [35,36]. This is suitable for our analysis since the outcome, diabetes, is a dichotomous variable. We used a hierarchical model to adjust for covariates sequentially in a stepwise manner, and thereby showed how adjustment for an increasing number and type of confounding variables affected the crude (unadjusted) estimates. Crude (unadjusted) and adjusted ORs (aORs) with 95% confidence intervals (CIs) were presented.

Demographic and lifestyle factors known or strongly suspected to increase the risk of diabetes (age, ethnicity, education level, menopausal status, smoking, alcohol consumption, physical activity, and height) [4,5,13,37,38] were adjusted for in Model 1. Since obesity is a well-known risk factor for type 2 diabetes [39], and visceral adiposity was previously found to be associated with insulin resistance in the IWHP cohort [40], VAT was added into Model 2. To adjust for muscle mass, ALM/height^2^, the normalized index recommended for use in Asian populations [27] was selected and added into Model 2. Model 3 added adjustment for HGS when assessing RCS, and adjustment for RCS when assessing HGS. 

We observed no association between pre-diabetes (fasting blood glucose 5.6–6.9 mmol/L [32]) and muscle strength: crude ORs (95% CIs) were 0.89 (0.52, 1.52) with low HGS and 1.04 (0.68, 1.59) with poor RCS. We therefore combined the small number (n = 94) of women with pre-diabetes with women without diabetes for all analyses.

Some data were missing for several important variables: ethnicity (3.5%), education level (1.2%), and physical activity (1.1%). To avoid the exclusion of participants who had one or more missing variables, we used multiple imputation by chained equations to impute missing values, with 20 imputations.

## 3. Results

### 3.1. Participant Characteristics

Of the 1201 women enrolled in this study, we excluded 11 with unknown diabetes status and 20 without RCS test data. In total, 1170 women (97.4%) were included. Those excluded tended to be nulliparous and had a normal MSI compared to those included (Appendix A). Among the 1170 women analyzed, 259 (22.1%) had weak HGS (<18 kg) while 519 (44.4%) performed poorly on the RCS test (≥12 s) (Figure 1). In total, 153 (13.1%) women had poor MSI (both HGS < 18 kg and RCS test ≥ 12 s) while 472 (40.3%) women had intermediate MSI (either HGS < 18 kg only or RCS test ≥ 12 s only). In total, 545 (46.6%) women had a normal MSI (both normal HGS (≥18 kg) and a good performance on the RCS test (<12 s)).

The average age was 56.3 ± 6.2 years, the majority were of Chinese ethnicity (83.5%), around two-thirds had highest education qualifications at pre-university level (65.0%), and 71.7% were postmenopausal (Table 1). An extremely low percentage of women were smokers (2.1%) or alcohol consumers (3.2%). More than half (53.7%) were overweight or obese. Diabetes was present in 12.1% of the participants.

There was a higher proportion of diabetic Malay and Indian women compared to non-diabetic women (Malays: 9.2% vs. 5.4%; Indians: 23.4% vs. 8.8%) (Table 1). Smokers and shorter women (<1.55 m) were more likely to be diabetic compared to non-smokers and taller women (≥1.55 m). Women with BMI ≥ 23.0 kg/m^2^ tended to be diabetic compared to women with BMI < 23.0 kg/m^2^. Among women with diabetes, 61.7% were in the highest VAT tertile (>131.0 cm^2^). Surprisingly, diabetic women tended to have increased muscle mass (68.8%) compared to non-diabetic women (45.5%).

Age, education level, housing type, parity, menopausal status, alcohol consumption, and physical activity were not significantly associated with diabetes status.

### 3.2. Association between HGS and Diabetes

A low HGS (<18 kg) was more prevalent in women with diabetes (31.2%) compared to women without (20.9%) (Table 1). Adjusted analyses revealed that women with weak HGS had a 1.59-fold (95% CI: 1.03, 2.44) higher odds of diabetes compared to those with a strong HGS (Table 2) after adjusting for age, ethnicity, education level, menopausal status, smoking, alcohol consumption, physical activity, height (Model 1), VAT, ALM/height^2^ (Model 2), and RCS (Model 3).

### 3.3. Association between RCS and Diabetes

A poor RCS performance (≥12 s) was more evident among women with diabetes (55.3%) compared to women without (42.9%) (Table 1). Poor performance on the RCS test was associated with 1.59-fold (95% CI: 1.09, 2.30) higher odds of diabetes. However, this association was attenuated upon adjustment for VAT, ALM/height^2^ (Model 2), and HGS (Model 3) (Table 2).

### 3.4. Association between Combined Muscle Strength Index (MSI) and Diabetes

Having a poor MSI was more prevalent among women with diabetes while an intermediate and normal MSI was more prevalent among women without diabetes (Table 1). In adjusted analyses, women with an intermediate MSI (weak HGS or prolonged RCS) were not associated with diabetes (Table 2). Consistent with our hypothesis, the poor MSI category (weak HGS and prolonged RCS) was associated with a 2.37-fold (95% CI: 1.40, 4.03) higher odds of diabetes even after sequential adjustment for the full set of covariates, including age, ethnicity, education level, menopausal status, smoking, alcohol consumption, physical activity, height, VAT, and muscle mass (ALM/height^2^).

## 4. Discussion

Our study supports the contention that muscle strength, independent of adiposity and muscle mass, is inversely associated with diabetes. We found that women with weak HGS were 1.59 times more likely to have diabetes. Our findings were consistent with that of a recent meta-analysis demonstrating a 27% reduced risk for diabetes in a comparison of the top vs. bottom thirds of HGS levels [13]. In adiposity-controlled models, it has been estimated that each SD increase in grip strength was associated with a 13% lower risk of diabetes [10]. Thresholds of normalized grip strength have been proposed as useful screening tools for diabetes risk in otherwise healthy adults [41].

Nevertheless, grip strength might not be entirely representative of whole-body strength since factors such as obesity might differentially affect the muscle strength of the upper and lower extremities. Our study suggests an association between lower body muscle strength and diabetes. Women with prolonged RCS test performance (some of whom also had weak HGS) were 1.59 times more likely to have diabetes, although the association was attenuated with adjustment for covariates, including HGS. To our knowledge, the RCS test has not been reported to be associated with diabetes among healthy subjects. A similar test is the Timed Up and Go test, which assesses fall risk, and involves rising from a chair, walking for 3 m, walking back to the chair, and sitting back down [42]. Older Chinese women who took a longer time on this test were reported to have higher risk of diabetes, possibly due to ageing-associated phenotypes such as muscle weakness and cognitive impairment [43].

We hypothesized that a composite muscle strength index combining both upper and lower body muscle strength would be better associated with diabetes than either HGS or the RCS test alone. In congruence with our hypothesis, we found that women with poor combined muscle strength were 2.37 times more likely to have diabetes compared to those with normal upper and lower body muscle strength, after adjusting for a wide range of covariates, including visceral adiposity. Participants with an intermediate MSI (either low HGS only or poor RCS test performance only) were not at risk for diabetes. Our findings confirm the results of a recent study in older Chinese subjects, which observed that a composite score incorporating both relative grip strength and the Timed Up and Go test was associated with incident diabetes [44]. Another study found that a moderate level of total body muscle strength, measured by the gold standard maximal weight an individual can lift for only one repetition, was associated with a lower risk of developing type 2 diabetes compared to those with lower muscular strength [45].

Various strands of evidence add to the growing recognition of the significant role of muscle strength in diabetes. In the Nurses’ Health Study, women who engaged in resistance training (at least 60 min/week of muscle-strengthening activities) had lower risks of diabetes compared to women who did not engage in such activities [46]. Resistance exercises increase muscle mass, the primary site for insulin-mediated glucose uptake (33), which enhances insulin sensitivity by increasing the GLUT4 protein content and insulin receptors [47]. Older adults with diabetes have lower leg muscle mass, strength, and quality than those without diabetes [48].

Our findings should be interpreted considering several limitations. Firstly, the cross-sectional design makes it difficult to infer causality, particularly because we cannot ensure that low muscle strength temporally preceded the onset of diabetes. Reduced muscle strength could theoretically have been a consequence of diabetes in our study women. Diabetic peripheral neuropathy, a common but severe complication of diabetes, might contribute to muscle strength decline [49]. Next, we did not collect data on the duration of diabetes, which might have affected its association with muscle strength. Longitudinal studies in this cohort are underway, and the use of incident diabetes rates would better examine this relationship with muscle strength. In addition, the chair stand test is not solely a measure of lower body muscle strength but also reflects balance, endurance, and fitness. Nonetheless, a moderately high correlation was found between the chair stand test and maximum weight-adjusted leg-press performance among older women (>60 years) [50]. Another limitation is possible residual confounding, which no observational study can exclude entirely. However, we accounted for many important covariates, including demographic factors and VAT. Finally, we were unable to differentiate between type 1 and type 2 diabetes, as we did not enquire about the type or age of onset of diabetes in our baseline questionnaire. Nonetheless, type 1 diabetes is uncommon in Singapore, and thus nearly all women identified with diabetes in our study are likely to have type 2 diabetes.

Strengths of our study include our use of the combined MSI, which accounts for both upper and lower body muscle strength, based on cut-offs from the recently updated Asian Working Group on Sarcopenia definition in 2019. Most previous studies examining muscle strength with diabetes were based on handgrip strength, whereas our study additionally assessed lower body strength via the RCS test.

## 5. Conclusions

In conclusion, we found that a low HGS was independently associated with diabetes status in midlife Singaporean women. A poor performance on the RCS test was associated with diabetes status as well, albeit not independently of visceral adiposity and muscle mass. Poor MSI, combining low upper and lower body strength, was strongly associated with diabetes. Future studies of muscle strength as a risk factor for diabetes should account for the muscle strength of both the upper and lower body. Longitudinal studies are required to inform the causal relationship and direction between muscle strength and diabetes.

## Figures and Tables

**Figure 1 ijerph-19-13654-f001:**
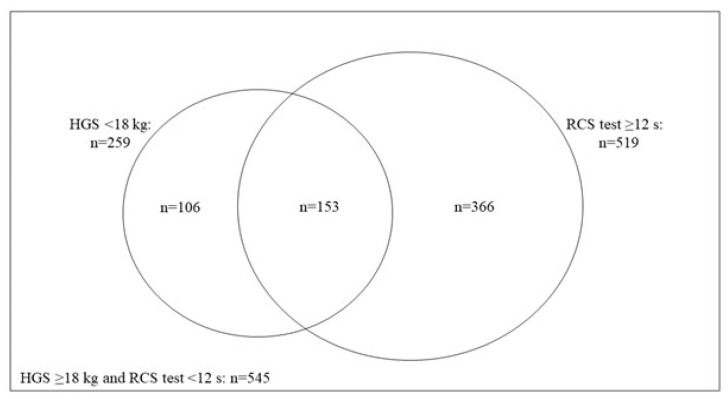
Venn diagram showing the distribution of muscle strength in 1170 subjects. Weak handgrip strength (HGS < 18 kg) only: n = 259 (106 + 153), prolonged 5-repetition chair stand test (RCS ≥ 12 s) only: n = 519 (153 + 366), muscle strength index (MSI): poor MSI (HGS < 18 kg and RCS ≥ 12 s): n = 153, intermediate MSI (HGS < 18 kg only, or RCS ≥ 12 s only): n = 472 (106 + 366), and normal MSI (HGS ≥ 18 kg and RCS < 12 s): n = 545 (1170 − 472 − 153).

**Table 1 ijerph-19-13654-t001:** Participant characteristics.

Characteristics	Total N (n = 1170)	No Diabetes (n = 1029, 87.9%)	Diabetes (n = 141, 12.1%)	*p*-Value
Self-reported measures, n (%)				
Age, years (mean ± SD)				0.491
45–54 (50.5 ± 2.7)	511 (43.7)	453 (44.0)	58 (41.1)	
55–64 (59.1 ± 2.9)	512 (43.8)	451 (43.8)	61 (43.3)	
65–69 (66.7 ± 1.3)	147 (12.6)	125 (12.1)	22 (15.6)	
Ethnicity				<0.001
Chinese	977 (83.5)	882 (85.7)	95 (67.4)	
Malay	69 (5.9)	56 (5.4)	13 (9.2)	
Indian	124 (10.6)	91 (8.8)	33 (23.4)	
Education level				0.232
Primary level or below	172 (14.7)	151 (14.7)	21 (14.9)	
Pre-university	761 (65.0)	662 (64.3)	99 (70.2)	
University	237 (20.3)	216 (21.0)	21 (14.9)	
Housing type				0.496
Public (1–3 room)	140 (12.0)	120 (11.7)	20 (14.2)	
Public (4–5 room)	795 (67.9)	705 (68.5)	90 (63.8)	
Private	235 (20.1)	204 (19.8)	31 (22.0)	
Parity				0.207
Nulliparous	197 (16.8)	168 (16.3)	29 (20.6)	
Multiparous	973 (83.2)	861 (83.7)	112 (79.4)	
Menopausal status				0.643
Premenopausal	147 (12.6)	132 (12.8)	15 (10.6)	
Perimenopausal	184 (15.7)	159 (15.5)	25 (17.7)	
Postmenopausal	839 (71.7)	738 (71.7)	101 (71.6)	
Smoking status				0.014
Non-smokers	1145 (97.9)	1011 (98.3)	134 (95.0)	
Smokers	25 (2.1)	18 (1.7)	7 (5.0)	
Alcohol consumption				0.075
Non-alcohol consumers	1133 (96.8)	993 (96.5)	140 (99.3)	
Alcohol consumers	37 (3.2)	36 (3.5)	1 (0.7)	
Physical activity				0.191
Yes	715 (61.1)	622 (60.4)	93 (66.0)	
No	455 (38.9)	407 (39.6)	48 (34.0)	
Objectively measured parameters, n (%)				
Height (m)				0.038
<1.55	418 (35.7)	354 (34.4)	64 (45.4)	
1.55–1.60	472 (40.3)	424 (41.2)	48 (34.0)	
>1.60	280 (23.9)	251 (24.4)	29 (20.6)	
BMI (kg/m^2^)				<0.001
Underweight (<18.5)	59 (5.0)	57 (5.5)	2 (1.4)	
Normal (18.5–22.9)	482 (41.2)	446 (43.3)	36 (25.5)	
Overweight (23.0–27.5)	403 (34.4)	352 (34.2)	51 (36.2)	
Obese (>27.5)	226 (19.3)	174 (16.9)	52 (36.9)	
Visceral adipose tissue (cm^2^)				<0.001
<88.6	393 (33.6)	379 (36.8)	14 (9.9)	
88.6–131.0	385 (32.9)	345 (33.5)	40 (28.4)	
>131.0	392 (33.5)	305 (29.6)	87 (61.7)	
ALM/height^2^				<0.001
<5.4	606 (51.8)	561 (54.5)	45 (31.9)	
≥5.4	564 (48.2)	468 (45.5)	96 (68.1)	
Handgrip strength (kg)				0.006
<18	259 (22.1)	215 (20.9)	44 (31.2)	
≥18	911 (77.9)	814 (79.1)	97 (68.8)	
5-repetition chair stand test (s) ^1^				0.005
≥12	519 (44.4)	441 (42.9)	78 (55.3)	
<12	651 (55.6)	588 (57.1)	63 (44.7)	
Muscle strength index				<0.001
Poor ^2^	153 (13.1)	119 (11.6)	34 (24.1)	
Intermediate ^3^	472 (40.3)	418 (40.6)	54 (38.3)	
Normal ^4^	545 (46.6)	492 (47.8)	53 (37.6)	

Multiple imputation was performed. ^1^ 5-repetition chair stand test: Time taken to rise from a seated position and back down to a sitting position five times. ^2^ Poor MSI: HGS < 18 kg and RCS ≥ 12 s. ^3^ Intermediate MSI: HGS < 18 kg or RCS ≥ 12 s. ^4^ Normal MSI: HGS ≥ 18 kg and RCS < 12 s.

**Table 2 ijerph-19-13654-t002:** Associations between HGS, RCS, and the muscle strength index (MSI) with diabetes status (n = 1170). Hierarchical binary logistic regression was performed with two categories (diabetes vs. non-diabetes). Results were presented as odds ratio (OR) and 95% confidence intervals (95% CIs), and significance indicated: * *p* < 0.05; ** *p* < 0.01; *** *p* < 0.001.

Muscle Strength Measures	Unadjusted	Model 1	Model 2	Model 3
	OR (95% CI)			
HGS (kg)				
<18	1.72 (1.17, 2.53) **	1.61 (1.07, 2.42) *	1.66 (1.09, 2.55) *	1.59 (1.03, 2.44) *
≥18	Reference			
RCS test (s)				
<12	Reference			
≥12	1.65 (1.16, 2.35) **	1.59 (1.09, 2.30) *	1.43 (0.98, 2.10)	1.36 (0.92, 2.00)
MSI				
Poor ^1^	2.65 (1.65, 4.27) ***	2.49 (1.50, 4.14) ***	2.37 (1.40, 4.03) **	-
Intermediate ^2^	1.20 (0.80, 1.79)	1.13 (0.75, 1.71)	1.04 (0.68, 1.60)	-
Normal ^3^	Reference			-

Analyses were performed after multiple imputation. Model 1: adjusted for age, ethnicity, education level, menopausal status, smoking, alcohol consumption, physical activity, and height. Model 2: adjusted for Model 1, VAT and ALM/height^2^. Model 3: adjusted for Model 2 and mutually for RCS and HGS, respectively. ^1^ Poor MSI: HGS < 18 kg and RCS ≥ 12 s. ^2^ Intermediate MSI: HGS < 18 kg or RCS ≥ 12 s. ^3^ Normal MSI: HGS ≥ 18 kg and RCS < 12 s.

## Data Availability

The data presented in this study are available on reasonable request from the corresponding author.

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
