# Peer review of "The Associations between Upper and Lower Body Muscle Strength and Diabetes among Midlife Women"

_ijerph, 2022, doi:10.3390/ijerph192013654_

Round 1

Reviewer 1 Report

Dear Authors, your Manuscript The associations between upper and lower body muscle strength and diabetes among midlife women is a well written and interesting article investigating the potential risk in low muscle strength and aquiering diabetes. You included  a respctable number of patients with well defined inclusion criteria. As you already acknowledged, the main limitation is not having the data on type and duration of diabetes and also, due to study desing, inability to detect causal relationship. It would be of importance also to investigate the association of muscle strength with prediabetes and if it affects progression to type 2 diabetes. In addition, conditions such as postural hypotension, diabetic polyneuropathy might alter the results of lower muscle strength. 

Author Response

Point 1: It would be of importance also to investigate the association of muscle strength with prediabetes and if it affects progression to type 2 diabetes.

Response 1: In our original submission, Methods, section 2.4, lines 129-131, we made an inadvertent typographical error.  It should read as follows:

“As we observed no association between pre-diabetes (fasting blood glucose 5.6-6.9 mmol/L [32]) and muscle strength (data not shown), we combined the small number (n=84) of women with pre-diabetes with women without diabetes.”

As our study has a cross-sectional design, we are unable to investigate associations with progression to type 2 diabetes from pre-diabetes. However, we plan to do so longitudinally in our follow-up study, which is well underway in our cohort.

Point 2: In addition, conditions such as postural hypotension, diabetic polyneuropathy might alter the results of lower muscle strength.

Response 2: We did not collect data on postural hypotension or diabetic polyneuropathy. However, these conditions are surely rare among our cohort, because 1) the mean age of participants at recruitment was 56 years of age, whereas diabetic polyneuropathy generally affects older persons with longstanding diabetes, and 2) we deliberately recruited healthy women.

Reviewer 2 Report

The authors tried to find the some possible connections between muscle strength and diabetes. However, they only investigated such association among women; men were not included. Meanwhile, the statistical analysis shown in this study was not enough to support their findings. This study needed further investigation and more data to support their conclusions. Some questions needed addressing.

1. Why did the authors want to investigate the association of muscle strength and diabetes in women? Whether such connection could be found in men?

2. What was the purpose of two indicators discussed by the authors? Whether these two could be used for predictors for diabetes screening?

3. Why did the authors employ hierarchical binary logistic regression? How did the analysis exclude the impact of other factors, such as smoking, alcohol drinking? The current evidence shown by authors did not fully support their conclusions. More statistical analysis could be provided.

Author Response

Point 1:  Why did the authors want to investigate the association of muscle strength and diabetes in women? Whether such connection could be found in men?

Response 1: We deliberately restricted our cohort to mid-life women and therefore excluded men. In our originally submitted manuscript, Introduction, lines 45-48, we stated:

“Muscle strength decline is more prominent among women than men after 55 years of age [11], possibly due to reduced muscle protein synthesis during the menopausal transition period [12]. Few studies have examined the contribution of declining muscle strength to diabetes in midlife women [10,13].”

Point 2:  What was the purpose of two indicators discussed by the authors? Whether these two could be used for predictors for diabetes screening?

Response 2: The reason for indicators of both upper and lower body strength were clearly summarized in the fourth paragraph of the Introduction of our original submission, along with relevant reference citations.   

We believe that our findings are far more relevant to understanding the causes of reduced muscle strength than to screening for diabetes, which is far better and more accurately accomplished with conventional blood glucose and glycated haemoglobin measurements.

Point 3 Why did the authors employ hierarchical binary logistic regression? How did the analysis exclude the impact of other factors, such as smoking, alcohol drinking? The current evidence shown by authors did not fully support their conclusions. More statistical analysis could be provided.

Response 3:  We used a hierarchical model to adjust for covariates sequentially and thereby show how adjustment for an increasing number and type of confounding variables affected the crude (unadjusted) estimates. This was explained in our original submission in Methods, lines 159-167:

“Demographic and lifestyle factors known to correlate with diabetes such as age, ethnicity, education level, menopausal status, smoking, physical activity, and height were chosen a priori [4,5,13,35] and entered as covariates into Model 1. Since obesity is a well-known risk factor for type 2 diabetes [36], and visceral adiposity was previously found to be correlated with insulin resistance in the IWHP cohort [37], VAT was added into Model 2. To adjust for muscle mass, ALM/height2, the normalised index recommended for use in Asian populations [27] was selected and added into Model 2. Model 3 added adjustment for HGS when assessing RCS, and adjustment for RCS when assessing HGS.

The effect of adjustment for smoking on the association between muscle strength and diabetes was stated in line 160 of our original submission. As suggested by the reviewer, we now include alcohol consumption in Model 1. As now mentioned in the Methods, lines 159-161:

“Demographic and lifestyle factors known or strongly suspected to increase the risk of diabetes (age, ethnicity, education level, menopausal status, smoking, alcohol consumption, physical activity, and height) [4,5,13,35,36] were adjusted for in Model 1.”

Round 2

Reviewer 2 Report

Some question was not clearly replied. The authors need to make improvement of contents to previous questions. More words should be provided about How the authors conduct the statistical analysis. Some of results about the statistical analysis were missed in the manuscript.

Q3: The authors didn't explain the principle of hierarchical binary logistic regression in the method. How to evaluate the model is good? For example, in table 2, what did the listed numbers mean? What was p value for each model?
